# Multiomic Approach to Analyze Infant Gut Microbiota: Experimental and Analytical Method Optimization

**DOI:** 10.3390/biom11070999

**Published:** 2021-07-07

**Authors:** Helena Torrell, Adrià Cereto-Massagué, Polina Kazakova, Lorena García, Héctor Palacios, Núria Canela

**Affiliations:** Eurecat, Centre Tecnològic de Catalunya, Centre for Omic Sciences (COS), Joint Unit Universitat Rovira i Virgili-EURECAT, Unique Scientific and Technical Infrastructures (ICTS), 43204 Reus, Spain; adria.cereto@ce.eurecat.org (A.C.-M.); polina.kazakova@eurecat.org (P.K.); lorena.garcia@eurecat.org (L.G.); hector.palacios@eurecat.org (H.P.); nuria.canela@eurecat.org (N.C.)

**Keywords:** next-generation sequencing, Ion Torrent, metagenomics, microbiome, early infancy microbiome, metabolomics, multiomics approach

## Abstract

Background: The human intestinal microbiome plays a central role in overall health status, especially in early life stages. 16S rRNA amplicon sequencing is used to profile its taxonomic composition; however, multiomic approaches have been proposed as the most accurate methods for study of the complexity of the gut microbiota. In this study, we propose an optimized method for bacterial diversity analysis that we validated and complemented with metabolomics by analyzing fecal samples. Methods: Forty-eight different analytical combinations regarding (1) 16S rRNA variable region sequencing, (2) a feature selection approach, and (3) taxonomy assignment methods were tested. A total of 18 infant fecal samples grouped depending on the type of feeding were analyzed by the proposed 16S rRNA workflow and by metabolomic analysis. Results: The results showed that the sole use of V4 region sequencing with ASV identification and VSEARCH for taxonomy assignment produced the most accurate results. The application of this workflow showed clear differences between fecal samples according to the type of feeding, which correlated with changes in the fecal metabolic profile. Conclusion: A multiomic approach using real fecal samples from 18 infants with different types of feeding demonstrated the effectiveness of the proposed 16S rRNA-amplicon sequencing workflow.

## 1. Introduction

The gut microbiota is the population of microorganisms (mainly bacteria but also viruses, protozoa, and fungi) and their collective genetic material present in the gastrointestinal tract. The intestinal microbiota plays an important role in human health, and the disruption of its composition, named dysbiosis, has been proposed as one of the most important factors involved in gastrointestinal diseases and other illnesses [1]. The gut microbiota is highly malleable and can be altered throughout the lifespan by environmental factors, such as diet and medications. The development of the microbiota begins at birth, when very low microbial diversity is observed in the gastrointestinal tract of newborns, with the microbial population dependent on the mode of delivery (cesarean section or vaginal delivery). Many studies have shown a correlation between the acquired microbiota during the first stages of life and the development of different diseases throughout life [2,3]. Pediatric intestinal disorders cover a wide range of injuries and conditions that impact a child’s intestines, being celiac disease (1%) [4], irritable bowel syndrome (9.3% to 35.5%) [5], and ulcerative colitis (0–3%) [6] the most relevant. Intestinal disorders can result from numerous conditions, but intestinal microbiota is thought to play an important role in illness development.

Complex transformations requiring microbial collaborations that are tightly regulated and coupled through microbial community interactions occur in the gut microbiota. There are several projects which aim to better our understanding of how the intestinal microbiota impacts human health and disease, one of the most important being The Human Microbiome Project (HMP) supported by the National Institute of Health (NIH).

The current trend in the application of omics pipelines in analyzing the actual status of diseases is evident [7,8,9]. Metagenomics is the study of a community of microorganisms by analyzing genomic sequences directly obtained from samples with no need to isolate and clone individual species [10]. The development of next-generation sequencing (NGS) techniques has allowed the production of high-quality and cost-effective genomic data, enough to identify and even relatively quantify microbial taxonomic units [11]. The common approach for identifying and classifying microbial species in a sample is to PCR amplify and further sequence a region (or several) of the 16S ribosomal RNA (16S rRNA) gene [12]. The use of different 16S rRNA gene regions means that the results of different studies are often not directly comparable, diminishing the value of the inferences that can be drawn. Biases in the detection of bacteria for a given region are known to be caused by the choice of primers and the amplification protocol [11,13,14,15,16,17,18,19,20].

After improvement of sequencing methods and the appearance of NGS [21], more sequences of metagenomic data from different samples were obtained. Consequently, many 16S rRNA gene databases have been established. The most important 16S databases are the Ribosomal Database Project, RDP [22], GreenGenes [23,24] Silva [25], and Eztaxon-e [26]. To accurately identify the microbial composition of metagenomic data, traditional approaches cluster sequences into operational taxonomic units (OTUs) [27]. There are several tools for OTU selection. One of the first tools was DOTUR, [28] but UCLUST and USEARCH [29] became the most commonly used tools in the last decade, and VSEARCH [30] has emerged as an open-source alternative to USEARCH. A more recent approach for taxonomic analysis is the use of amplicon sequence variants (ASVs). Instead of clustering sequences by similarity, all unique sequences are retained after filtering low-quality and erroneous reads, which allows for far greater resolution for detecting different yet very similar sequences. ASV methods are thus able to resolve sequence differences by as little as a single nucleotide change, which allows this method to avoid similarity-based operational clustering units all together.

There are several bioinformatic tools for the identification of taxonomic classifications of metagenomes, each of which works with algorithms and pipelines. MG-RAST [31] is an online tool for annotation and taxonomic classification of metagenomes. ESPRIT [32] is an algorithm that removes low-quality reads, computes pairwise distances, groups OTUs and calculates statistical inferences to estimate the species richness [33]. EMIRGE [34] is a program for sequencing short reads of Illumina, and it is an iterative method to assemble small subunit ribosomal genes (SSUs) of metagenomic data and estimate relative taxa abundance [35]. Two of the most commonly used programs are MOTHUR [36] and Quantitative Insights Into Microbial Ecology (QIIME) [37,38]. The pipelines of these programs are similar, with their main difference being in scope; one is a single program reimplementing everything (MOTHUR), and the other is a collection of scripts leveraging already existing tools, where one feeds to the next in a chain of inputs and outputs, which gives it access to a wider range of algorithms.

Due to the wide range of analytical possibilities, we aimed to select an optimized and validated workflow to assess bacterial diversity by comparing the results with a known bacterial community comprised of eleven species. The relative performance characteristics to be compared included employing two distinct methods of 16S rRNA library preparation (custom fusion-tag primers and a commercial kit named Ion 16S Metagenomics kit from Life Technologies) and several informatic configurations regarding feature selection and taxonomy assignment when using QIIME2 [39]. We selected QIIME2 due to its customizable input and output options and to enable increased transparency, reproducibility, and the use of open-source methods.

Once the optimized workflow was identified, it was validated through fecal samples from 18 infants with different types of feeding (breastfed and formula-fed) in the first month of life. Additionally, as recent multiomic approaches have been proposed as the most accurate methods for the study of the complexity of gut microbiota functions [9], 16S rRNA metagenome analysis was complemented with fecal metabolome profiling by nuclear magnetic resonance (NMR).

## 2. Materials and Methods

### 2.1. Method Optimization

#### 2.1.1. Bacterial Strains and DNA Preparation

Eleven representative microorganisms were used in this study. These strains include *Lactobacillus brevis* 216, *Pediococcus parvulus* 3911, *Lactobacillus plantarum* 220, *Pediococcus pentosaceus* 4208, *Lactobacillus hilgardii* 4786, *Bacteroides coprophilus*, *Prevotella copri*, *Gluconobacter oxydans*, *Acetobacter malorum* 14377, *Lactobacillus buchnerii* 4111T, and *Escherichia coli* DBH10.

The genomic bacterial DNA from pure cultures was obtained by extraction with the QIAamp DNA stool kit (Qiagen, Hilden, Germany) following the protocol for Isolation of DNA from Stool for Pathogen detection (with slight modifications). The bacterial pellet or the stool samples were homogenized with 2 mL of lysis buffer ASL. Incubation at 96 °C was performed for 10 min unaided by any combination of digestion enzymes or detergents to break the membranes of both Gram+ and Gram- bacteria. Then, the manufacturer’s protocol was followed for the rest of the steps. The final elution was performed with 200 µL of nuclease-free water. DNA quantification was performed with a Nanodrop2000 spectrophotometer (ThermoFisher Scientific, MA, USA).

#### 2.1.2. Library Preparation

Two different methods were used for amplifying the selected 16S rRNA gene regions.

##### Tag-Fusion Primers

Partial 16S rRNA gene sequences were amplified from the extracted DNA using primer pair 341F-532R, which targets the V3 region of the 16S rRNA gene sequence, primer pair 515F-806R, which targets the V4 region, and primer pair 967F-1046R, which targets the V6 region (Table 1). The coordinates are based on the 16S rRNA gene of *Escherichia coli* strain K-12 substr. MG1655.

Primer selection (Table 1) was performed by searching the scientific literature [40,41,42], assessing their specificity using BLAST tools, and ensuring that they matched conserved regions of the 16S rRNA gene [43].

These primers were designed to include at their 5′ end one of the two adaptor sequences used in the Ion Torrent sequencing library preparation protocol, linking a unique tag barcode of 10 bases to identify different samples (Table 1).

The PCR cycle parameters used for the V4 and V6 regions were 3 min at 94 °C, 25 cycles of 30 s at 94 °C, 45 s at 57 °C and 60 s at 72 °C, followed by 2 min at 72 °C. For the V3 region, the thermal cycle was 5 min at 94 °C, 25 cycles of 30 s at 94 °C, 45 s at 55 °C and 60 s at 72 °C followed by 10 min at 72 °C. The amplification conditions employed were 18.6 µL HiFi Platinum (Life Technologies, Carlsbad, CA, USA), 1.5 µL of each primer at 2.5 µM, and 50 ng of genomic DNA in a final volume of 25 µL. Reactions were carried out by using a Verity Thermocycler (Applied Biosystems, Foster City, CA, USA). PCR products were confirmed by a 2% agarose gel, and specific bands were excised and then purified using a Nucleospin Gel and PCR clean up kit (Macherey-Nagel, Berlin, Germany), dissolving the gel slices at room temperature by vortexing (without heat). The concentration of the PCR amplicons was analyzed by electrophoresis on a Bioanalyzer (Agilent Technologies, Santa Clara, CA, USA). Equimolar pools of each fragment were combined, and 26 pM equimolar pools of all strains were also formed.

##### Ion 16s Metagenomic Kit from Life Technologies

Seven regions of the 16S rRNA gene were amplified with an Ion 16s Metagenomics kit (Life Technologies, Carlsbad, CA, USA), purified using AgentCourt AMPure beads (Beckman Coulter, Brea, CA, USA) and quantified using a Universal Library Quantitation kit (Life Technologies, Carlsbad, CA, USA). All procedures were performed following the manufacturer’s instructions. These procedures included amplification of regions V2, V3, V4, V6, V7, V8, and V9 by two primer pools, ligation of the specific Ion Torrent adaptors and barcodes, nick repair, and final DNA purification.

#### 2.1.3. Ion Torrent PGM Sequencing

Once the libraries were created, they were diluted to 26 pM DNA prior to clonal amplification. Emulsion PCR and ion sphere particle enrichment were carried out using the Ion PGM Template OT2 Hi-Q Kit (Life Technologies, Carlsbad, CA, USA) according to the manufacturer’s instructions. Afterward, the samples were prepared for sequencing by employing the Ion PGM Hi-Q Sequencing Kit (Life Technologies, Carlsbad, CA, USA). Prepared samples were loaded on a 318 chip and sequenced using the Ion Torrent PGM applying 840 flows.

After sequencing, the individual sequence reads were filtered by PGM software to remove low-quality and polyclonal sequences. Those reads were then converted to FASTQ files and analyzed using QIIME2.

#### 2.1.4. QIIME2 Analysis

The sequences were divided according to the library preparation method: kit (Ion 16s Metagenomic kit from Life Technologies, Carlsbad, CA, USA) and tag-fusion primers, which were further divided according to the amplified region: V3, V4, and V6. Combinations of the data from the latter were also analyzed (V3 + V4, V3 + V6, V4 + V6, and V3 + V4 + V6). All of the regions included in the commercial 16S metagenomics kit were analyzed together.

From this point, several configurations were tried, with every possible combination of the following parameters:

Feature Selection approach: Recent amplicon sequence variant (ASV) approach using dada2 [44] or the traditional operational taxonomic unit (OTU) picking method using VSEARCH [29] with 99% homology. Quality control retained sequences for which all bases had a quality score of 20 or higher.

Taxonomy assigning method: The default consensus alignment was conducted with VSEARCH, BLAST [45], or a recently proposed newer approach through a sklearn machine learning method (Naïve Bayes). The database used for both the consensus alignment reference and to train the naïve Bayes model was the Silva database [46] at 99% homology (which has been shown to outperform the 97% that is widely used [47].

All these possibilities yielded a total of 48 different combinations (Figure 1).

#### 2.1.5. Performance Comparison

To compare the performance of each method in terms of accuracy with respect to the actual sample composition, there is the need for a metric that can evaluate each method’s performance. This metric should be able to compare the difference between the expected results and the obtained results in terms of percentage abundance. We chose to use Bray–Curtis dissimilarity to compare the conditions. A perfect analysis method would yield 9.09% of reads identified for each of the 11 species and 0% of the remaining reads, which would give a Bray–Curtis dissimilarity of 0 for the sample composition. A method absolutely incapable of identifying any read as one of the expected species would yield a dissimilarity of 1. Thus, the closer the dissimilarity is to 0, the closer the result’s analysis is to the actual composition of the sample. The further its absolute value increases, the further the analysis’s results deviate from reality.

The performance of all methods was compared at three different taxonomic levels: family, genus, and species. The best workflow was selected according to its lower dissimilarity value. Additionally, the performance of ASV identification versus OTU picking regardless of library construction method was statistically analyzed using the Holm–Šídák post hoc adjustment of the Kruskal–Wallis test.

### 2.2. Method Validation

To confirm the workflow capacity to describe bacterial diversity in gut microbiota, the optimized procedure was applied to 18 infant fecal samples. The infants were under six months old and exclusively fed milk. The infants were classified depending on the type of feeding (breast feeding, *n* = 9 or formula feeding, *n* = 9). Additionally, the fecal metabolic profile was also analyzed through NMR in 15 of the previous 18 fecal samples (breast feeding, *n* = 9, and formula feeding, *n* = 6). A multiomic correlation was performed to relate the metabolite changes with gut microbiota differences in composition (Figure 2). Samples were stored at −80 °C until metagenomic and metabolic analysis.

#### 2.2.1. Fecal Bacterial Diversity Analysis through Partial 16S rRNA Sequencing

DNA was extracted from approximately 200 mg of fecal sample using a QIAamp^®^ DNA Stool Mini Kit (Qiagen Inc., Hilden, Germany) according to the manufacturer’s instructions. Library preparation, sequencing, and taxonomy assignment were performed as described in Section 3.1. The analysis included amplicon sequence variant (ASV) identification, taxonomy assignment, and alpha-diversity analysis (Shannon index). Taxonomic abundances were compared between experimental groups using the Holm–Šídák post hoc adjustment of the Kruskal–Wallis test.

#### 2.2.2. Fecal Metabolite Profiling by NMR

Fecal samples (50 mg) were homogenized with phosphate-buffered saline (PBS). The homogenates were centrifuged at 15,000 g for 15 min at 4 °C. Two-hundred microliters of the supernatants were separated from the pellet and mixed with 400 µL of D_2_O phosphate buffer (PBS 0.05 mM, pH 7.4, 99.5% D_2_O). The mixture was transferred into 5 mm NMR glass tubes for NMR measurement. The NMR spectra were measured at a 600.20 MHz frequency while using an Avance III-600 Bruker spectrometer equipped with a 5 mm PABBO BB-1H/D Z-GRD probe. A standard ^1^H NOESY presaturation pulse sequence (RD-90°-t1-90°-tm-90°-acquire, noesypr1d) was used with water suppression. A recycle delay (RD) of 5.0 s, a mixing time (tm) of 100 ms, an acquisition time of 3.4 s, and a 90° pulse of 21.16 µs were used for all of the samples. NMR spectra were processed using TopSpin 3.5pl4 software (Bruker Biospin, Conventry, UK). An exponential line broadening of 0.3 Hz was applied before Fourier transformation. Metabolite identification was carried out using information from the literature, public databases, and proprietary software (Chenomx NMR Suite 8.5^®^, Human Metabolite DataBase, Biological Magnetic Resonance Data Bank), and normalization was performed using the PULCON (PULse length-based CONcentration determination) methodology (ERETIC^®^), which is based on the principle of reciprocity. Orthogonal partial least squares-discriminant analysis (OPLS-DA) was performed with Metaboanalyst software 5.0 (https://www.metaboanalyst.ca/; accessed on 23 October 2020). The statistical calculation of the significance of the correlated OPLS-DA metabolites (*p* < 0.05) was obtained using the website for statistical computation VassarStats (http://www.vassarstats.net/; accessed on 27 October 2020).

#### 2.2.3. Integration between Metabolites and Gut Bacterial Composition

Metabolite levels and microbial diversity were analyzed through neural networks with MMVECs (Metabolite-Microbe VECtor) [48] to test whether there was co-occurrence between the type of feeding, the obtained metabolites through NMR and the detected bacterial taxonomic categories. This was achieved by training the neural network to predict metabolite compositions based solely on the presence of each individual microbe and then inferring co-occurrence probabilities from the trained model and the taxonomic profile of each sample.

## 3. Results

### 3.1. Method Optimization

The 16S rRNA gene NGS run produced 5,141,956 reads from the two library preparation protocols. These reads were transformed to FASTQ format, and two FASTQ files were used as input in the QIIME2 analysis workflows. All results were split depending on the 16S rRNA gene region used to assign taxonomy prior to the QIIME2 analysis. Analyzing the dissimilarity data in detail, they clearly showed that use of the V4 region applying the ASV approach for feature selection and VSEARCH for taxonomy assignment produces the most accurate results at the three studied taxonomic levels (Table 2). However, the differences between this combination and the combination of ASV identification with BLAST or sklearn for taxonomy assignment is minimal at the genus and family levels (Figure 3). Thus, ASV identification performed better than OTU picking at the family and genus levels regardless of the taxonomy assignment method (*p* = 0.001). However, at the species level, OTU picking + VSEARCH showed significantly lower dissimilarity values than the other combinations involving ASV identification (corrected *p* < 0.05, Figure 3), although these results were conditioned by the 16S rRNA gene region considered (Table 2). On the other hand, the incorporation of the V6 region into the analysis causes high dissimilarity values due to its impossibility of assigning the *Bacteroidaceae* family (data not shown). Thus, the exclusive use of the V3 and V4 regions seems more suitable as the dissimilarity values are smaller. However, the combination of V3 + V4 regions did not increase the performance of the analysis, showing that the amplification of only the V4 region was sufficient to correctly assess the taxonomy, at least at the genus and family levels.

Notably, when comparing the expected and obtained results (Table 3) per taxonomic level, it is clear that they are almost the same at the family and genus levels but not at the species level with six of the species not identified. These data also confirm that the addition of V3 region sequences does not improve the taxonomy assignment accuracy.

### 3.2. Method Validation

#### 3.2.1. Differences in Bacterial Diversity between the Experimental Groups

The sequencing run generated 5,423,214 reads that were used to determine ASVs from the V4 region of the 16S rRNA gene, which were then used to summarize the relative abundance of the microbial clades at different taxonomic levels. These differences are mainly due to changes in the Firmicutes and Bacteroidetes phyla (predominant in the breast-fed group (44% and 30%) and Proteobacteria (predominant in the formula-fed group (57%)). These differences were also transferred to lower taxonomic levels (Figure 4a). A significant decrease in the *Enterobacteriaceae* family (*p* = 0.0064) was observed in the breast-fed group compared to the formula-fed group, paired with an increase in the *Staphylococcaceae* (*p* = 0.016), *Porticoccaceae* (*p* = 0.022), and *Immundisolibacteraceae* (*p* = 0.022) families. Additionally, greater species richness was observed in the samples belonging to the breastfed group, although the differences were not significant (Figure 4b).

#### 3.2.2. Multivariate Analysis Showed Differences in the Fecal Metabolome Depending on the Type of Feeding

A total of 27 metabolites were identified and quantified after the alignment and normalization of the spectra. Table 4 shows the mean concentration per each metabolite per group, and additionally raw data has been upload to MetaboLights (ebi.ac.uk/metabolights; accessed on: 6 June 2021) with the accession number MTBLS2942. To analyze the differences between the two groups, a pairwise OPLS-DA model was performed to compare the effect of lactation on the fecal metabolome. A model with predictive ability [R2X = 0.218; Q2 = 0.307] was obtained comparing the fecal metabolites of both groups. The metabolites that were significantly altered (*p* < 0.05) in this model and found to be higher in the formula-fed group were valine, phenylalanine, lactate, isoleucine, glycine, citrate, choline, aspartate, alanine, and 2-hydroxy-3-methylbutyric. We also observed tendencies in different metabolites, such as an increase in butyrate concentration (*p* = 0.078) and isoleucine (*p* = 0.077) in the formula-fed group. No significant metabolites were found to be increased in breastfed infants compared with the formula-fed group (Figure 5) [49].

#### 3.2.3. Integration between Metabolites and Gut Microbiome

Some of the analyzed metabolites co-occurred with the bacterial population in the infant fecal samples. The most relevant co-occurrences were observed in aspartate with a positive correlation with the *Staphylococcales*, *Clostridiales*, *Lachnospirales*, and *Pasteurellales* orders; lactate with a negative correlation in the *Clostridiales* order; and glycine with a positive correlation in the *Bifidobacteriales* order and a negative correlation with the *Lachnospirales* order (Figure 6).

## 4. Discussion

### 4.1. Method Optimization

These results showed that the analysis of the V4 region using ASV identification and VSEARCH for taxonomy assignment produced the most accurate results at the three studied taxonomic levels.

These results demonstrated that the precise assessment of the composition of a given microbiota population depends on several items, either methodological or analytical. Different choices applied to the same sample may lead to different outcomes in describing the microbial population. Here, we used an Ion Torrent Machine to generate genomic data. It has been documented that Ion Torrent exhibits a higher rate of sequencing errors than data from the Illumina platform [50]; however, in partial 16S rRNA sequencing approaches, both platforms are generally comparable [51].

Regardless of the sequencing platform, the first aspect to consider is the selection of 16S rRNA gene regions to be sequenced. We examined the V3, V4, and V6 regions and analyzed the complete 16S rRNA gene amplification mediated by a 16S metagenomics kit from Life Technologies to compare the accuracy of all library preparation approaches. This commercial kit allows the sequencing of several regions of the 16S rRNA gene, but in our study, it did not lead to the expected results. One explanation for this discordance can be erroneous primer design, varying amplification efficiencies and coverage rates of the primers contained in this kit (unknown sequences), or high error and chimera rates [52]. These results do not agree with a previous study that also analyzed the performance of the Thermo Fisher 16S kit compared with a commercial service based on V4 sequencing and concluded that the Ion 16S kit generally allowed the greatest number of taxonomic identifications across the domain Bacteria. However, it used different bioinformatic pipelines to assess taxonomy [20]. Nevertheless, we insist on considering cost and convenience factors. For instance, tag-fusion primers are easy to design and cheap to acquire when working on the Ion Torrent platform (depending on how many samples are multiplexed in a single run and how many variable regions are chosen). Furthermore, library construction through tag-fusion primers requires less hands-on time and fewer steps than commercial kits. In addition, the amplification of solely one variable region per sample permits more samples per run, as more reads can be generated per sample without losing resolution.

For variable region selection, V3 and V4 are currently the most popular regions to be examined, although V6 has also traditionally been used [40,53,54]. We observed that the V6 region led to inaccurate results, suggesting insufficient specificity for that region or for the designed primers, perhaps due to polymorphisms accumulated in the conserved regions or perhaps due to its short length. Some studies have suggested that the full sequencing of the V3–V4 regions provides the closest analog to sequencing the entire 16S rRNA gene [55], while others have proposed that the single use of the V4 region is sufficient [56,57]. However, all of them applied OTU picking for feature selection instead of ASV identification. In our study, although the combination of the V3 and V4 regions generated more reads, the sequencing information obtained from the V4 region alone was able to assign the correct taxonomic rank to all of the identified ASVs at the genus level. Regardless of the informatic workflow applied, our approach failed in assigning taxonomy at the species level, probably due to the low taxonomic and phylogenetic resolution of the 16S rRNA gene analyzed regions. The reasons behind these peculiarities may be related to the functional variable regions: the V4, V5, and V6 regions directly take part in translation and are responsible for binding tRNAs and interacting with the 23S rRNA; therefore, these regions should be the most conserved. It is logical to think that more conserved regions should be sufficiently distinct only at higher taxa levels, while less conserved regions could distinguish among the lower levels [55]. To solve this issue, in experiments where it is highly important to determine the bacterial population at the species level, it has been proposed to additionally sequence the rRNA internal transcribed spacer (ITS) [58]. This taxonomic profiling can be performed using tag-fusion primers and can be analyzed on QIIME because specific ITS databases are available.

Regarding feature selection approaches [21], we applied QIIME’s open-reference OTU picking method using VSEARCH, which also used a quality filter and checked for chimeras in the input reads before OTU picking. We also performed ASV identification, for which QIIME offers two possible backends: DADA2 and deblur [59]. Per QIIME2 documentation, deblur should be used only on Illumina reads, and as our reads were single-end reads generated with a different technology, we selected DADA2 instead. Notably, OTU approaches have served the microbiome community for many years and will likely still find use in the future in specific circumstances; however, there is an ever-increasing amount of evidence that ASV approaches are the superior choice for most future analyses [60]. Thus, in this study, ASVs were expected to outperform OTU-based approaches, and they did so at the family and genus levels. The unexpected result at the species level, with no consistent winner between OTUs and ASVs across different pipelines, may be due to the previous inability to obtain species-level resolution with this dataset [61]. A recent study also showed that DADA2 offered the best sensitivity as compared to other bioinformatic pipelines for the analysis of amplicon sequence data and concluded that ASV-level workflows offer superior resolution compared to OTU-level workflows [61].

The next step involves assigning taxonomies to the features (OTUs and ASVs), which again can be done with several different methods. QIIME’s available consensus alignment methods, using VSEARCH and BLAST, respectively, were tested, alongside the machine learning taxonomy assignment method based on a naïve Bayes model trained on the same reference database (Silva) [48]. Considering the results, the VSEARCH consensus alignment provides more reliable and consistent results, topping the results for all pipeline combinations. However, the difference from the other 2 methods is very small in most cases, while the sklearn-based method had the widest dispersion in performance. Similar results were shown in a previous study in which a sklearn naïve Bayes machine-learning classifier, alignment-based taxonomy consensus methods based on VSEARCH, and BLAST met or exceeded the species-level accuracy of other commonly used methods [62].

### 4.2. Method Validation through a Multiomics Approach

With the optimized workflow, and only selecting the V4 region of the 16S rRNA gene, it was possible to characterize the infant gut microbiota in fecal samples and separate them into two groups depending on the type of feeding, replicating already published studies. The small sample size of this study (*n* = 18 for microbiota analysis and *n* = 15 for metabolomic profile) must be noted. Although it is the main limitation; this sample size is enough for an exploratory study. A significant decrease in the *Enterobacteriaceae* family (Mann-Whitney, *p* = 0.0064) was observed in the breast-fed group compared to the formula-fed group, paired with an increase in the *Staphylococcaceae*, *Porticoccaceae*, and *Immundisolibacteraceae* families. These results agree with previous studies comparing the influence of the type of feeding on gut microbiota in infants and when using a similar workflow with only the V4 region of the 16S rRNA gene with the QIIME 1.8 and Greengenes reference databases [63].

Additionally, the multiomic approach, with the inclusion of metabolic profiling from feces, contributes to separating the two groups and is also correlated with changes in the microbiota. We observed tendencies in different metabolites, such as an increase in butyrate concentration and isoleucine in the formula-fed group, as reported previously [49]. Levels of choline were higher in feces of the formula-fed group in contrast with a previous study where the choline concentration was higher in breast-fed infants, but they were using serum samples [64]. These results might be due to high levels of these metabolites in the formula milk administered to infants. The observed positive correlation between bacteria from the Firmicutes phylum and butyrate may be explained by its ability to produce this short-chain fatty acid (SCFA) [65,66]. The negative correlation between *Bifidobacteriales* and citrate and other SCFAs that we observed was also described previously [67]. It is important to note that intrinsic factors such as genetics and other environmental factors such as lifestyle may be influencing both microbiota and metabolic phenotypes, and thus contributing to differentiate both experimental groups.

## 5. Conclusions

It has been shown that microbiota community description profiling can be affected by the specific target region used, library preparation method and analysis workflow because all of them will give different results in terms of error rates and biases. We concluded that the sequencing of the V4 region only, using ASV identification for feature selection and VSEARCH for taxonomy assignment, produces the most accurate results when working on the Ion Torrent NGS platform. However, all combinations struggled to correctly identify the samples at the species level, but the results up to the genus level were satisfactory. Notably, this setup outperformed the use of the commercial primer kit.

The optimized workflow applied to fecal samples in an infant nutrition study demonstrated its ability to assign taxonomy and to discern between samples and groups when working with a fecal matrix. There is a need for workflow standardization when analyzing bacterial diversity through 16S rRNA gene partial sequencing, and here we provide a validated protocol which can be applied either in research, clinical or commercial field to accurate determine bacterial communities from a variety of samples. Furthermore, its complementation with other omic data, such as metabolite profiling, increases the potential use of metagenomics to enlarge the knowledge of microbiota functions and relationships with host physiopathology.

## Figures and Tables

**Figure 1 biomolecules-11-00999-f001:**
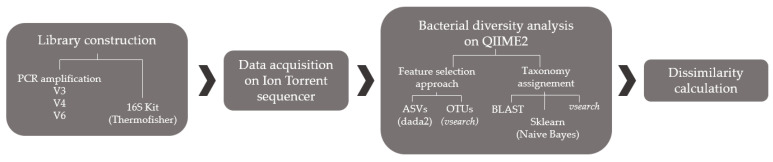
Scheme of all of the possible analytical combinations.

**Figure 2 biomolecules-11-00999-f002:**
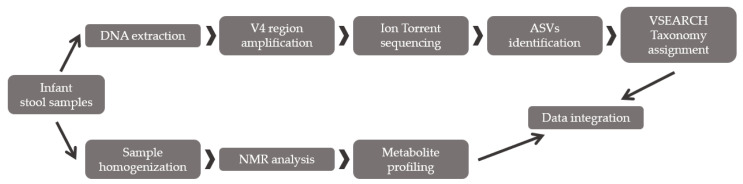
Scheme of the applied multiomic approach: bacterial diversity analysis combined with fecal metabolite profiling.

**Figure 3 biomolecules-11-00999-f003:**
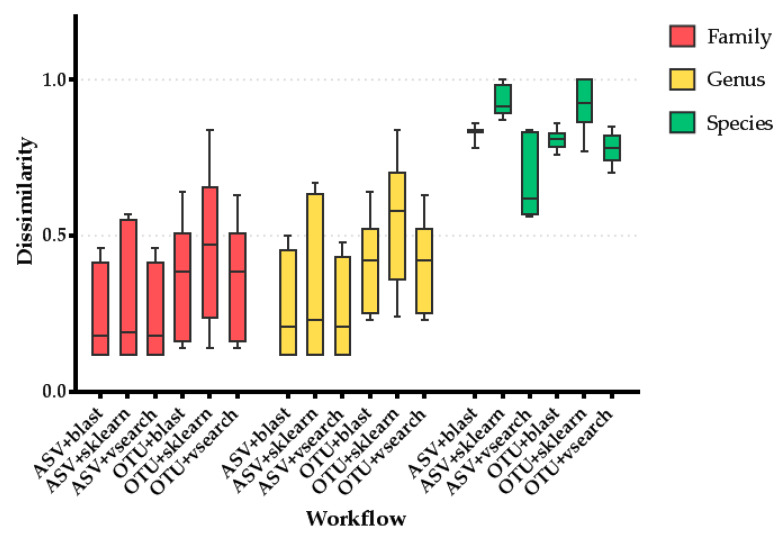
This graph shows the dissimilarity to the actual composition for the combinations of the QIIME feature selection approach and taxonomy assignment method at three different taxonomic levels. The following comparisons showed statistically significant differences at the species level (corrected *p* < 0.05): ASV+sklearn vs. ASV + VSEARCH; ASV+sklearn vs. OTU+blast; ASV + sklearn vs. OTU + VSEARCH; ASV + VSEARCH vs. OTU + VSEARCH; OTU + sklearn vs. OTU + VSEARCH.

**Figure 4 biomolecules-11-00999-f004:**
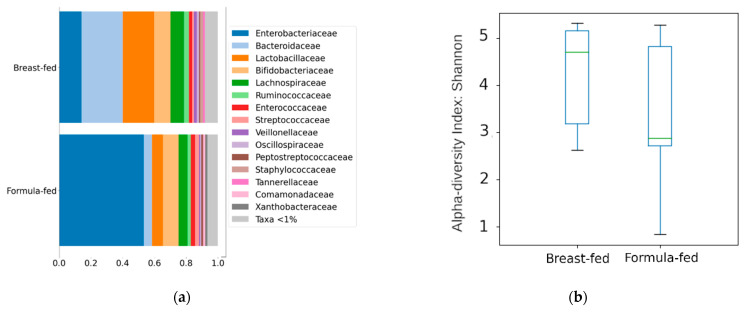
(**a**) Taxonomic composition of the community at the Family level using a Stacked Bar plot. (**b**) Alpha diversity measure using Shannon, Y axis represent Shannon score values.

**Figure 5 biomolecules-11-00999-f005:**
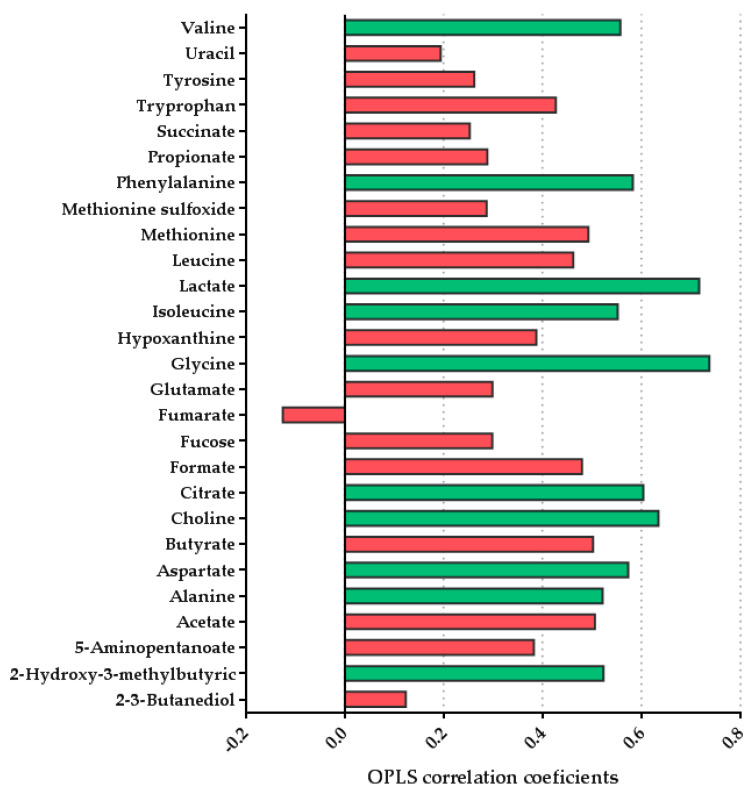
OPLS-DA line plot indicating metabolite differentiation between the groups (*n* = 9 breast-fed group and *n* = 6 formula-fed group). Positive correlation coefficients indicate higher levels in the formula-fed group. The significant metabolite bars are represented in green (*p* < 0.05).

**Figure 6 biomolecules-11-00999-f006:**
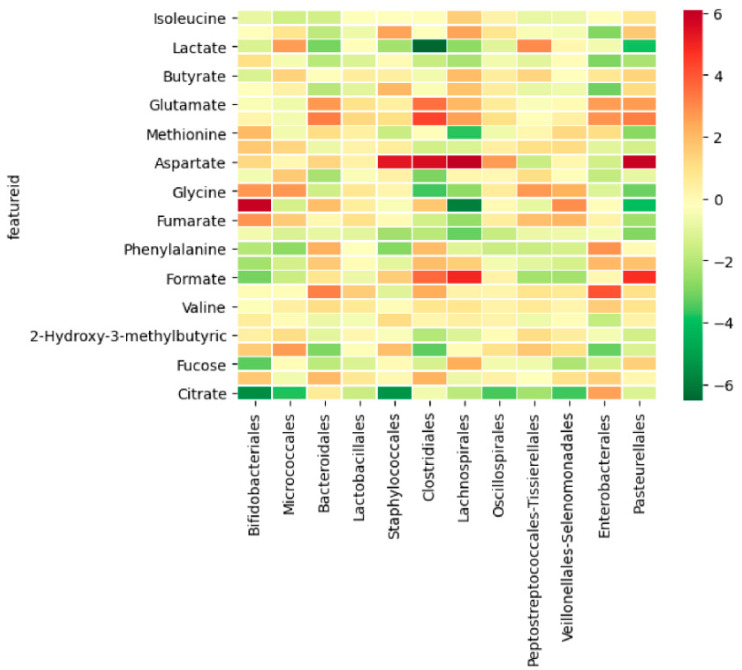
Heatmap with correlation co-occurrence between metabolites and the bacterial community at the order level. Darker red shows a stronger positive co-ocurrence (both occur together); dark green shows a stronger negative co-ocurrence (one occurs in the absence of the other one); pale yellow shows no significant co-ocurrence.

**Table 1 biomolecules-11-00999-t001:** Fusion-tag primer sequences used to amplify the V3, V4, and V6 16S rRNA gene regions.

Name and Direction	Adaptor	Key	Barcode	Spacer	Target 16S Primer
V3_forward	CCATCTCATCCCTGCGTGTCTCCGAC	TCAG	CTAAGGTAAC	GAT	CCTACGGGRSGCAGCAG
V3_reverse	CCTCTCTATGGGCAGTCGGTGAT			CC	ATTACCGCGGCTGCT
V4_forward	CCATCTCATCCCTGCGTGTCTCCGAC	TCAG	CTAAGGTAAC	GAT	GTGCCAGCMGCCGCGGTAA
V4_reverse	CCTCTCTATGGGCAGTCGGTGAT			CC	GGACTACHVGGGTWTCTAAT
V6_forward	CCATCTCATCCCTGCGTGTCTCCGAC	TCAG	CTAAGGTAAC	GAT	CAACGCGAAGAACCTTACC
V6_reverse	CCTCTCTATGGGCAGTCGGTGAT			CC	CGACAGCCATGCANCACCT

**Table 2 biomolecules-11-00999-t002:** This table shows the dissimilarity values obtained at three taxonomic levels (family, genus and species), depending on the library construction method, the feature selection approach and the taxonomy assignment method.

Library Construction	Feature Selection	Taxonomy	Bray–Curtis Dissimilarity		
			Species	Genus	Family
V4	ASV	VSEARCH	0.5576	0.1182	0.1182
V3_V4	ASV	VSEARCH	0.5577	0.1182	0.1182
V3	ASV	VSEARCH	0.5930	0.1186	0.1186
V3_V4_V6	ASV	VSEARCH	0.6221	0.2084	0.1756
V4_V6	ASV	VSEARCH	0.6222	0.2087	0.1758
V3	OTU	VSEARCH	0.6966	0.2342	0.1405
V3_V6	OTU	VSEARCH	0.7268	0.2617	0.1903
V3_V6	OTU	BLAST	0.7627	0.2601	0.1916
V3	OTU	sklearn	0.7695	0.2387	0.1445
kit	OTU	VSEARCH	0.7712	0.2506	0.1532
V6	OTU	BLAST	0.7770	0.3955	0.3580
V3_V4	OTU	VSEARCH	0.7791	0.4680	0.4374
kit	ASV	BLAST	0.7792	0.3138	0.3137
V3_V4_V6	OTU	VSEARCH	0.7825	0.4392	0.4086
V3	OTU	BLAST	0.7872	0.2338	0.1405
V6	OTU	VSEARCH	0.7916	0.3961	0.3562
kit	ASV	VSEARCH	0.7978	0.3138	0.3137
kit	OTU	BLAST	0.8077	0.2477	0.1529
V3_V4_V6	OTU	BLAST	0.8088	0.4386	0.4096
V3_V4	OTU	BLAST	0.8208	0.4682	0.4378
V4_V6	OTU	VSEARCH	0.8262	0.5376	0.5264
V4	ASV	BLAST	0.8305	0.1182	0.1182
V3_V4	ASV	BLAST	0.8307	0.1182	0.1182
V4_V6	OTU	BLAST	0.8312	0.5371	0.5278
V4_V6	ASV	BLAST	0.8351	0.2088	0.1759
V3_V4_V6	ASV	BLAST	0.8351	0.2085	0.1757
V3_V6	ASV	VSEARCH	0.8413	0.4719	0.4536
V6	ASV	VSEARCH	0.8438	0.4761	0.4577
V6	ASV	BLAST	0.8443	0.5015	0.4584
V3_V6	ASV	BLAST	0.8445	0.4970	0.4543
V4	OTU	VSEARCH	0.8454	0.6350	0.6346
kit	OTU	sklearn	0.8568	0.3560	0.2626
V4	OTU	BLAST	0.8614	0.6355	0.6353
V3	ASV	BLAST	0.8631	0.1186	0.1186
kit	ASV	sklearn	0.8724	0.5185	0.5181
V3_V6	OTU	sklearn	0.8756	0.3614	0.2275
V3	ASV	sklearn	0.8914	0.1186	0.1186
V3_V4	ASV	sklearn	0.8972	0.1182	0.1182
V4	ASV	sklearn	0.8973	0.1182	0.1182
V3_V4	OTU	sklearn	0.9144	0.6126	0.5830
V3_V4_V6	ASV	sklearn	0.9282	0.2321	0.1864
V4_V6	ASV	sklearn	0.9284	0.2325	0.1869
V3_V4_V6	OTU	sklearn	0.9367	0.5926	0.5302
V3_V6	ASV	sklearn	0.9958	0.6680	0.5621
V6	OTU	sklearn	0.9958	0.5720	0.4090
V6	ASV	sklearn	0.9968	0.6740	0.5672
V4_V6	OTU	sklearn	0.9985	0.7347	0.6812
V4	OTU	sklearn	1.0000	0.8434	0.8434

**Table 3 biomolecules-11-00999-t003:** This table shows the expected relative taxonomic abundance from the synthetic sample (Expected) and the relative abundance reported (Obtained) by the ASV + VSEARCH analysis with the V3, V4 and combination of both regions at different taxonomic resolutions.

Species	Expected	Obtained V3 + V4	Obtained V3	Obtained V4
*Lactobacillus brevis*	9.09%	8.80%	6.62%	8.80%
*Pediococcus pentosaceus*	9.09%	10.05%	8.83%	10.06%
*Lactobacillus plantarum*	9.09%	11.89%	11.48%	11.89%
*Gluconobacter oxydans*	9.09%	8.14%	7.06%	8.14%
*Bacteroides coprophilus*	9.09%	0.00%	0.00%	0.00%
*Lactobacillus hilgardii*	9.09%	0.02%	0.00%	0.02%
*Escherichia coli*	9.09%	9.99%	9.93%	9.99%
*Prevotella copri*	9.09%	0.00%	0.00%	0.00%
*Pediococcus parvulus*	9.09%	0.00%	0.00%	0.00%
*Acetobacter malorum*	9.09%	0.00%	0.00%	0.00%
*Lactobacillus buchneri*	9.09%	0.00%	0.00%	0.00%
Other	0	51.11%	56.07%	51.09%
**Genus**	**Expected**	**Obtained** **V3 + V4**	**Obtained** **V3**	**Obtained** **V4**
*Bacteroides*	9.09%	9.33%	10.30%	9.32%
*Prevotella*	9.09%	7.63%	8.68%	7.62%
*Lactobacillus*	36.36%	37.08%	36.72%	37.08%
*Pediococcus*	18.18%	18.96%	18.54%	18.96%
*Acetobacter*	9.09%	8.76%	8.76%	8.76%
*Gluconobacter*	9.09%	8.14%	7.06%	8.15%
*Escherichia*	9.09%	9.99%	9.93%	9.99%
Other	0.00%	0.10%	0.00%	0.10%
**Family**	**Expected**	**Obtained** **V3 + V4**	**Obtained** **V3**	**Obtained** **V4**
*Bacteroidaceae*	9.09%	9.33%	10.30%	9.32%
*Prevotellaceae*	9.09%	7.63%	8.68%	7.62%
*Lactobacillaceae*	54.55%	56.04%	55.26%	56.04%
*Acetobacteraceae*	18.18%	16.90%	15.82%	16.91%
*Enterobacteriaceae*	9.09%	9.99%	9.93%	9.99%
Other	0	0.10%	0.00%	0.10%

**Table 4 biomolecules-11-00999-t004:** Mean concentration per group in mmol/mg of representative fecal metabolites analysed by Nuclear Magnetic Resonance. Desvest means standard deviation and SEM stands for standard error of the mean.

	Breast-Fed Group			Formula-Fed Group		
Metabolite	Mean	Desvest	SEM	Mean	Desvest	SEM
Valine	1.25	0.76	0.25	3.51	2.43	0.99
Uracil	0.05	0.06	0.02	0.14	0.08	0.04
Tyrosine	0.12	0.08	0.03	0.16	0.09	0.04
Tryprophan	0.06	0.05	0.02	0.11	0.06	0.02
Succinate	2.12	2.12	0.71	6.17	10.47	4.28
Propionate	1.73	1.24	0.41	2.33	1.23	0.50
Phenylalanine	0.12	0.07	0.02	0.30	0.17	0.07
Methionine sulfoxide	0.03	0.03	0.01	0.04	0.04	0.02
Methionine	0.14	0.08	0.03	0.33	0.20	0.08
Leucine	0.69	0.39	0.13	1.58	1.05	0.43
Lactate	0.48	0.37	0.12	7.82	8.12	3.32
Isoleucine	0.79	0.51	0.17	1.96	1.17	0.48
Hypoxanthine	0.06	0.03	0.01	0.10	0.07	0.03
Glycine	0.41	0.18	0.06	2.61	2.17	0.89
Glutamate	0.83	0.71	0.24	1.48	1.08	0.44
Fumarate	0.03	0.02	0.01	0.02	0.01	0.00
Fucose	0.21	0.30	0.10	0.25	0.17	0.07
Formate	0.02	0.01	0.00	0.07	0.07	0.03
Citrate	0.18	0.15	0.05	1.18	1.15	0.47
Choline	0.07	0.04	0.01	0.17	0.06	0.02
Butyrate	1.21	1.24	0.41	3.99	2.78	1.13
Aspartate	0.21	0.12	0.04	0.67	0.70	0.29
Alanine	1.31	0.86	0.29	3.24	2.06	0.84
Acetate	6.72	5.71	1.90	16.64	12.19	4.98
5-Aminopentanoate	0.82	1.70	0.57	1.22	1.41	0.58
2-Hydroxy-3-methylbutyric	0.04	0.03	0.01	0.11	0.07	0.03
2-3-Butanediol	0.23	0.16	0.05	0.44	0.60	0.25

## Data Availability

Metagenomic data analyzed in the study are available upon request to the authors of the article. Metabolomic data can be found in https://www.ebi.ac.uk/metabolights/ (accessed on 6 June 2021) under the accession number MTBLS2942.

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
