# Peer review of "Multiomic Approach to Analyze Infant Gut Microbiota: Experimental and Analytical Method Optimization"

_biomolecules, 2021, doi:10.3390/biom11070999_

Round 1

Reviewer 1 Report

In this work, infant microbiote for bacterial diversity was analysed.

A metabolomics workflow was suggested and used to the fecal sample types in the number of 18, working on the basis to be classified according to feeding differences.

Different fecal samples based on the feeding conditions, the differences were observed on the RNA level.

REMARKS

INTRODUCTION (note, 1st paragraph)

Include some current statistics associated with problems of gut microbiota issues such as diet or pharmaceuticals applications and others, from which infants would suffer. The reader would be interested more about the current status and numbers of gut microbiota problems in infants in different ground world areas. Also, please name the organisations that deal with statistics regarding this serious problem.

INTRODUCTION (note, 3rd paragraph)

Please, follow the upgrade citing all mentioned important papers dealing with recent developments in –omics approaches for different purposes, for example medical or clinical ones.

INTRODUCTION (uprade, 3rd paragraph)

The current trends in application of –omics pipelines in analysing of actual statues of diseases is evident [[https://www.sciencedirect.com/science/article/pii/S0959437X20301581, https://www.sciencedirect.com/science/article/pii/S0958166918302386, https://www.sciencedirect.com/science/article/pii/S0009898120301704]. Metagenomics is the study of a community of microorganisms by analyzing genomic sequences directly obtained from samples with no need to isolate and clone individual species [4]. The development of next-generation sequencing (NGS) techniques has allowed the production of high-quality and cost-effective genomic data, enough to identify and even relatively quantify microbial taxonomic units [5].

MATERIALS AND METHODS (note)

Please, draw the proposed analytical protocol to display particular steps from sampling to analysis.

CONCLUSION (note)

Please, indicate some of your future aims in this area of investigation and also what would be the possibilities to use this protocol to be applied in the clinical research on the commercial level.

Author Response

We thank the reviewer for the usefull comments.

According to the suggestions, the introduction has been updated regarding child intestinal disorders. We included some current statistics and mentioned an important world organization that deals with them (line 40). We also included some references regarding multiomics approaches when studying intestinal microbiota role in health and disease (line 50).

In materials and methods section, we included a scheme representing the followed procedure to analyze stool samples. It comprises from sampling to multiomics analysis and has been named “figure 2” (line 206).

Additionally, in conclusions section we now emphasize the importance of a standardized protocol and the usefulness of the proposed workflow to accurately characterize bacterial population through partial sequencing of 16S rRNA gene (line 429).

Reviewer 2 Report

In this study, Torrell et al used multiomic approach, e.g. 16S rRNA variable region sequencing and metabolomic profiling to analyze the infant gut microbiota. The results suggest differences between fecal samples according to the type of feeding. The work is ambitious and have the potential to make impact in the field. However, due to the complexity of the data, extra caution should be taken. In the metabolomic profiling by NMR, it is very challenging to accurately assign the metabolites and even more so to integrate the peaks given the complexity of 1D spectra. The authors should provide the raw data and related analyses to show the confidence of the peak assignment and integration. 

Some minor points: 
1. The sample size 18 is rather small to make very I understand the difficulty to obtain more samples in a short time period, but the authors should at least acknowledge the limitation of the work. 

2. Could other factors contribute to the differences in the differences between fecal samples? Such as genetics? The authors should at least consider other possibilities. 

3. Fig. 3b, what's the Y axis? 

4. Fig. 4, how many times were the experiments repeated?

5. The authors should provide more explanations to Fig. 5 for laymen like me, e.g. the methodology used in here and what the positive and negative correlations mean in this case. 

6. What kind of NMR spectra were collected for metabolomic profiling? 

Round 2

Reviewer 1 Report

Authors have contributed to all given remarks. In this way, I consider publication of manuscript at current state

This manuscript is a resubmission of an earlier submission. The following is a list of the peer review reports and author responses from that submission.

Round 1

Reviewer 1 Report

This is an interesting study
where the authors propose a improved approach to better profile the taxonomic composition of the real fecal samples microbiota.
Workflow is well planned and described.

Since the popularity of microbiota research today,
this paper will make a significant scientific contribution.

Reviewer 2 Report

Dr. Torrell and colleagues performed an interesting study in order to compare different methods for the study of the gut microbiota and to find out the most performing ones. They showed method optimisation, validation and confirmation with fecal samples from infants fed with human milk or formula. The study is well conducted and presented, but I have some minor comments that need to be fixed before publication.

The introduction is way too long. It needs to be significantly shortened and focussed. The main objectives of the study should be clearly presented at the end of the introduction.

In the result section 3.1, the authors present the comparison between methods showing median values and interquartile ranges, however it is not clear if they performed statistical analyses (i.e. they do not present p values).

There are some typos throughout the manuscript.

Reviewer 3 Report

There are plenty of grammatical mistakes in the manuscript, e.g., OUT instead of OTU, and that's too in the abstract.

Species Identification with 16s rRNA is challenging, and the author should mention how much confidence the pipeline has in species identification and how many species are identified. Does any species is unidentified by the pipeline?

Why QIIME v 1.9. That's a very old version and has multiple issues. The author should use the QIIME 2 for any analysis. Also, if the study is about algorithm comparison, they should include a comparison of multiple algorithms. A number of studies have performed the comparison of different methods and they were missing in the references.

Also, the algorithm performance varies on the sequencing platform. The algorithm performance on Ion torrent data, which is not used much nowadays, would be very different on Illumina data, which is widely used nowadays.

The writing is tough to understand, e.g., the Distance calculation section is difficult to understand. e.g., "The perfect analysis method would yield 9,09% of reads identified for each of the 11228 species, and 0% of remaining reads,
2.1.4.3. Distance calculation". What is 9,09%? This paragraph is the core of the manuscript and is the most difficult one to understand.

The study is missing the reference "http://dx.doi.org/10.1093/bioinformatics/bty113" and many more. They should be discussed and why the author didn't perform at ~100%.